# Adaptive RAG-Assisted MRI Platform (ARAMP) for Brain Metastasis Detection and Reporting: A Retrospective Evaluation Using Post-Contrast T1-Weighted Imaging

**DOI:** 10.3390/bioengineering12070698

**Published:** 2025-06-26

**Authors:** Kuo-Chen Wu, Fatt Yang Chew, Kang-Lun Cheng, Wu-Chung Shen, Pei-Chun Yeh, Chia-Hung Kao, Wan-Yuo Guo, Shih-Sheng Chang

**Affiliations:** 1Artificial Intelligence Center, China Medical University Hospital, Taichung 40447, Taiwan; robinsixrainbow@gmail.com (K.-C.W.); 008252@tool.caaumed.org.tw (P.-C.Y.); dr.kaochiahung@gmail.com (C.-H.K.); wanyuoguo2014@gmail.com (W.-Y.G.); 2Department of Medical Imaging, China Medical University Hospital, Taichung 40447, Taiwan; 032208@tool.caaumed.org.tw (F.Y.C.); 023077@tool.caaumed.org.tw (K.-L.C.); 004805@tool.caaumed.org.tw (W.-C.S.); 3Department of Nuclear Medicine and PET Center, China Medical University Hospital, Taichung 40447, Taiwan; 4Department of Bioinformatics and Medical Engineering, Asia University, Taichung 41354, Taiwan; 5Graduate Institute of Biomedical Sciences, School of Medicine, College of Medicine, China Medical University, Taichung 40447, Taiwan; 6Division of Cardiovascular Medicine, Department of Medicine, China Medical University Hospital, School of Medicine, China Medical University, Taichung 40447, Taiwan; 7School of Medicine, China Medical University, Taichung 40447, Taiwan

**Keywords:** MRI, brain metastases, radiological, retrieval-augmented generation, axial T1-weighted

## Abstract

This study aimed to develop and evaluate an AI-driven platform, the Adaptive RAG Assistant MRI Platform (ARAMP), for assisting in the diagnosis and reporting of brain metastases using post-contrast axial T1-weighted (AX_T1+C) MRI. In this retrospective study, 2447 cancer patients who underwent MRI between 2010 and 2022 were screened. A subset of 100 randomized patients with confirmed brain metastases and 100 matched non-cancer controls were selected for evaluation. ARAMP integrates quantitative radiomic feature extraction with an adaptive Retrieval-Augmented Generation (RAG) framework based on a large language model (LLM, GPT-4o), incorporating five authoritative medical references. Three board-certified neuroradiologists and an independent LLM (Gemini 2.0 Pro) assessed ARAMP performance. Metrics of the assessment included Pre-/Post-Trained Inference Difference, Inter-Inference Agreement, and Sensitivity. Post-training, ARAMP achieved a mean Inference Similarity score of 67.45%. Inter-Inference Agreement among radiologists averaged 30.20% (*p* = 0.01). Sensitivity for brain metastasis detection improved from 0.84 (pre-training) to 0.98 (post-training). ARAMP also showed improved reliability in identifying brain metastases as the primary diagnosis post-RAG integration. This adaptive RAG-based framework may improve diagnostic efficiency and standardization in radiological workflows.

## 1. Introduction

Brain metastasis management represents a complex clinical challenge. Brain metastases affect approximately 20–40% of cancer patients and contribute substantially to morbidity and mortality [1]. These metastases, which develop when cancer cells spread from primary tumors to the brain tissue, often indicate advanced disease stages and are associated with limited therapeutic options. The presence of brain metastases necessitates early and accurate diagnosis to guide clinical decision-making, treatment strategy selection, and ultimately improve patient outcomes [2,3]. Magnetic resonance imaging (MRI) remains the gold standard for detecting brain metastases, owing to its superior sensitivity in soft tissue differentiation and high spatial resolution for detecting small lesions [4]. However, despite continuous technological advances in imaging acquisition and processing, diagnostic workflows continue to face substantial challenges in both efficiency and reliability.

Current clinical imaging diagnostic practice relies heavily on radiologists’ interpretation of complex and variable imaging presentations, a task that demands considerable expertise and time. Radiologists must carefully evaluate multiple imaging series, assess various lesion characteristics on MRI, and give a diagnosis or multiple differential diagnoses. Imaging interpretation is expertise-dependent and inherently subjective, with inter-interpreter variability. The complex and lengthy imaging diagnosis process may potentially delay treatment initiation and compromise patient care and clinical outcomes [5]. Furthermore, the increasing volume of medical imaging data, coupled with the growing complexity of diseases, creates substantial workflow bottlenecks in clinical practice. All of the above exacerbate a common challenge in the global medical community: the shortage of radiologists. These limitations underscore the pressing need for innovative imaging diagnostic tools that can enhance both accuracy and efficiency in clinical workflows while maintaining consistent diagnostic standards.

Artificial intelligence (AI) and radiomics have emerged as promising solutions to address these challenges. AI technologies, particularly deep learning algorithms, enable automated tumor detection, segmentation, and classification, facilitating the process of reading complex imaging data with remarkable speed and consistency. Radiomics complements these capabilities by extracting quantitative imaging features—including texture, shape, and intensity characteristics—to distinguish malignant from benign lesions and predict their clinical outcome with greater precision [6,7]. These approaches can potentially identify subtle imaging biomarkers that might escape human detection. Nevertheless, clinical implementation of these technologies faces several critical obstacles, including the need for standardized data acquisition protocols, limited model interpretability, concerns about generalizability across different patient populations, imaging protocols, and imaging scanners, and stringent validation requirements to ensure clinical safety and efficacy [8,9,10,11].

Retrieval-Augmented Generation (RAG) frameworks represent a recent advancement in radiology that potentially overcomes several limitations of conventional AI systems [12]. RAG incorporates external medical knowledge bases into AI models, enhancing diagnostic precision by leveraging established clinical expertise and enabling real-time retrieval of relevant clinical data. This sophisticated approach effectively addresses various challenges, including bias mitigation in model predictions, reduction of interpretation variability, and improvement of AI system transparency, while supporting personalized diagnostics based on individual patient characteristics and clinical contexts [13]. However, RAG systems face a significant limitation known as “hallucination,” where the generated outputs may contain inaccurate or unverifiable information derived from imperfect knowledge integration or retrieval processes. This limitation is particularly crucial in medical applications, where erroneous outputs can directly impact patient safety and clinical outcomes. To address these concerns, researchers have developed domain-specific RAG models designed to enhance accuracy and reliability in medical contexts through specialized knowledge bases and rigorous validation protocols [14].

In recent years, applying large language models (LLMs) such as GPT-4o and Google Gemini in the medical field has garnered widespread attention, particularly in evaluating their performance in professional examinations. Research indicates that these models demonstrate strong capabilities when processing text-based medical knowledge questions. For example, a study by [15] found that GPT-4o achieved a high score of 82.1% on text-only multiple-response questions (focusing on recall and understanding) from the European Diploma in Radiology (EDiR) examination, significantly outperforming actual candidates (radiologists, average 49.4%), especially excelling in identifying correct answers (high true positive rate).

However, their performance presents a different picture when evaluating LLMs’ ability to handle complex medical questions requiring visual input [15]. Ref [16] evaluated the capabilities of GPT-4o and Gemini regarding image-based questions from a neurosurgery board examination question bank. The results showed that GPT-4o’s accuracy (51.45%) surpassed Gemini’s (39.58%), particularly on second-order questions requiring diagnostic reasoning, and performed better in subspecialties like pathology/histology and radiology [16]. Notably, the study also found that the presence or absence of images had impacts on the models’ accuracy, suggesting that the models rely more on the textual context of the questions rather than genuinely analyzing the image content to arrive at answers.

While LLMs have reached or even surpassed human expert levels in handling text-based medical knowledge recall, their capabilities remain significantly limited in tasks requiring the integration of image interpretation and complex clinical reasoning. This indicates that image analysis and multimodal integration capabilities are key challenges for future development of multi-modal AI in the medical community.

In this study, we present a novel brain MRI Assistant platform that leverages adaptive RAG technology, namely Adaptive RAG-Assisted MRI Platform (ARAMP), for the diagnosis and reporting of brain metastases. This platform integrates advanced feature extraction for analyzing post-contrast T1-weighted axial view (AX_T1+C) MRI images with adaptive learning mechanisms, enabling seamless incorporation of multimodal data. Through real-time retrieval of relevant knowledge and contextualized generation of diagnostic insights, the system addresses key challenges in neuroradiology, such as differentiating metastases from other disease conditions and reducing diagnostic variability. The primary objective of this study was to develop and evaluate the performance of ARAMP in detecting brain metastases on AX_T1+C brain MRI images and generating imaging reports. To ensure transparency and reproducibility, all items in the CLAIM 2024 Checklist were addressed and documented in Appendix A. This includes details on the study design, data preprocessing, radiologist consensus, evaluation metrics, and clinical implications.

## 2. Materials and Methods

### 2.1. Study Population

This retrospective study was approved by the Institutional Review Board of China Medical University Hospital (CMUH) (IRB No. CMUH111-REC1-206). The requirement for informed consent was waived due to the retrospective design and the use of de-identified patient data.

This retrospective study utilized data from patients who underwent brain MRI examinations at our institution between 2010 and 2022. Patient records were electronically queried and cross-referenced with the Major Illness Database (MajorIllnessDim) and the Cancer Registry Database (CaRecord) to identify individuals with a confirmed cancer diagnosis. Inclusion criteria were (1) age greater than or equal to 20 years and (2) a confirmed cancer diagnosis within 60 days prior to the MRI examination or within one year after the examination. Patients not meeting these criteria were excluded from the study. The selection was stratified by cancer type to reflect the distribution observed in the overall cohort.

Detailed inclusion and exclusion criteria, as well as sampling procedures, are documented in Appendix A (Items 8, 10, 21, and 35).

From this cohort, we randomly selected 100 cases for in-depth analysis, ensuring that the sample reflected the original cohort in terms of cancer type, age, and gender distribution.

The sampling was stratified by primary cancer diagnosis, with matched age and sex distribution to enhance representativeness (see Table 1 and Flowchart in Appendix A). To evaluate the performance of our large language model (LLM) model in identifying potential indicators of brain metastases by post-contrast trans-axial T1-wieghted (AX_T1+C) MRI with intravenous administration of a gadolinium-based contrast agent, a non-cancer control cohort was also established. This cohort consisted of 100 individuals randomly selected from a pool of patients who had undergone MRI examination containing AX_T1+C imaging but did not have a documented cancer diagnosis within the specified timeframe. This sampling design allowed balanced radiomic feature comparison and enabled controlled input for GPT-4o inference. All patient DICOM datasets were de-identified and processed through a standardized radiomics pipeline, generating both numerical features and image composites (10-slice PNG format), as illustrated in Appendix A and the workflow diagram in Appendix A.

These control subjects were matched to the cancer cohort in terms of age and gender distribution to minimize potential confounding factors and ensure baseline comparability for subsequent analyses. The matching process involved replicating the proportional distribution of age and gender observed in the cancer patient cohort within the control cohort.

### 2.2. Subgroup Analysis

To further investigate the relationship between imaging features and the presence of brain metastases, we conducted subgroup analyses using the same stratified subsample of 100 cancer patients previously described in Section 2.1. This group was randomly selected from the initial cohort of 2447 patients, with age and gender distributions matched to the broader cancer population to ensure representativeness. Among the 100 patients, 52 were male, and 48 were female. The age distribution was balanced across gender and spanned a wide range, with the largest groups falling between the ages of 40 and 70 years old.

This stratified sampling approach was designed to preserve representativeness and minimize demographic bias, as required by CLAIM Items 8 and 10. Detailed metadata, including age, gender, and diagnosis codes, were cross-referenced using the institutional cancer registry. The clinical cancer diagnoses within this subsample were diverse, with lung cancer comprising the majority (*n* = 65). Other cancer types included breast cancer (*n* = 11), colorectal cancer (*n* = 3), esophageal carcinoma (*n* = 3), and various other cancers, such as melanoma, non-Hodgkin’s disease, and prostate cancer. This composition reflects national epidemiological trends for brain metastases in East Asia, ensuring external relevance of the test set (see Appendix A, CLAIM Item 22). The complete demographic and clinical distribution used for radiomic analysis and GPT-4o inference input is listed in Table 1. The full distribution of age, gender, and clinical diagnoses is detailed in Table 1.

For each patient in this subsample, the corresponding MRI images and reports were carefully reviewed by three board-certified radiologists, two with over 8 years and more than 40 years of experience in neuroradiology, respectively, to determine if there was any documented evidence of tumor metastasis (Mets). This manual review served as the gold standard for comparison with the LLM model’s predictions. To ensure consistency and reduce interpretive variability, the review protocol was standardized across cases and supported by structured report templates. All radiologist ratings were recorded in Appendix A.

#### Inter-Rater Reliability Analysis

To evaluate the effectiveness of model training and the stability of its outputs, two intraclass correlation coefficient (ICC) analyses were conducted using a two-way mixed effects model (Type C, consistency), based on physician ratings using a 5-point Likert scale. Scores were converted to a 0–100% scale for normalization and comparative analysis. Full rating matrices for pre- and post-training outputs, as well as Gemini-generated responses, are detailed in Appendix A. Corresponding CLAIM items are addressed in Appendix A (Items 36, 37, and 43).

### 2.3. Data Preparation for Radiomic and Multimodal Analysis

#### 2.3.1. Data Preprocessing

All directories containing DICOM files within the institutional database were systematically identified using the PyDicom version 3.0 function. This function recursively scanned the designated storage locations to locate all folders containing DICOM images. Subsequently, PyDicom functions were employed to filter and retrieve DICOM files specifically related to the AX_T1+C MRI. These functions utilized predefined criteria to identify and extract only the relevant DICOM files from the larger pool of imaging data, establishing a comprehensive list of file paths for subsequent processing. After selection, all DICOM files underwent de-identification and were processed using PyRadiomics. Images were resampled to isotropic 1 mm^3^ voxels using B-spline interpolation, and intensity normalization was applied (scale = 100, voxel shift = 1000). Feature extraction included original, LoG (σ = 3, 5 mm), and wavelet (Haar) transformations, with discretization bin width set to 25. These parameters are detailed in Appendix A.

#### 2.3.2. Feature Extraction

Individual DICOM image files were loaded and processed using PyDicom, a specialized library for handling DICOM image data. This process involved extracting the image pixel data, which represents the raw intensity values for each pixel in the image. Feature extraction was then performed using the PyRadiomics library’s feature extractor module. This module provides a standardized and validated framework for extracting a wide range of quantitative imaging features from medical images.

In this study, parameters within the feature extractor module were configured to enable extraction of four specific categories of imaging features:(1)First-order statistical features: These features capture basic intensity distribution characteristics within the image, such as mean, standard deviation, skewness, and kurtosis.(2)Shape features: These features quantify the geometric properties of the region of interest (ROI), including volume, surface area, sphericity, and maximum diameter.(3)LoG features: These features are based on the Laplacian of Gaussian (LoG) filter, which enhances edges and boundaries within the image. LoG features capture the texture and heterogeneity of the ROI.(4)Wavelet features: These features decompose the image into different frequency subbands, capturing texture information at multiple scales.

To ensure cross-patient comparability, all images were spatially resampled to an iso-tropic voxel spacing of [1, 1, 1] mm using B-spline interpolation (interpolator: sitkBSpline). Intensity normalization was applied with a voxel array shift of 1000 and scaling factor of 100. Laplacian of Gaussian (LoG) features were extracted using σ values of 3 and 5 mm, while wavelet decomposition was performed using Haar basis functions at level 1. The bin width for gray level discretization was set to 25. These parameter settings were applied uniformly to all images and are summarized in Appendix A.

Features were calculated for each patient’s image data in conjunction with a hypothetical label mask, which represents the region of interest (ROI) within the image. These extracted features were then organized and stored in a pandas DataFrame, a tabular data structure suitable for further analysis and machine learning applications.

#### 2.3.3. Image Enhancement and Normalization

To optimize the visual representation of the images and mitigate the impact of potential noise or artifacts, image values were enhanced and normalized by utilizing the 1st and 99th percentiles of the image minimum and maximum values. This approach effectively compressed the numerical range of the image data to a standardized interval of [0, 1], ensuring that the images were represented within a consistent and comparable scale.

This percentile-based contrast stretching was applied prior to PNG image generation to ensure consistent grayscale representation for subsequent GPT-4o multimodal inference.

All enhanced images were saved in de-identified PNG format, with 10 contiguous slices combined into each composite image. This transformation pipeline is illustrated in Appendix A.

The complete set of enhancement parameters, image normalization strategy, and saving conventions used in this study were aligned with preprocessing standards defined in CLAIM 2024 (Appendix A, Items 13 and 24).

#### 2.3.4. DICOM-to-PNG Conversion

Following the enhancement and normalization procedures described in Section 2.3.3, the processed DICOM images were subsequently converted into the more widely compatible PNG format to facilitate visualization and downstream model input. This conversion process involved sequentially arranging the DICOM images, with a maximum of 10 layers combined into a single PNG image. The arrangement within the PNG followed a two rows by five columns matrix structure, effectively organizing the image layers into a visually coherent grid. OpenCV, a comprehensive library for image processing and computer vision tasks, was used to save these converted images in PNG format.

This layout design ensures consistency across inputs and enables rapid visual inspection of brain lesion morphology by both human reviewers and GPT-4o’s vision encoder.

The PNG images were de-identified and stored alongside structured metadata (e.g., radiomic features, patient ID, and axial location) in a standardized folder structure to support reproducibility and linkage with inference outputs.

The complete image conversion and arrangement process is depicted in Appendix A and complies with CLAIM 2024 standards for image format compatibility and interpretability (Appendix A, Item 24).

### 2.4. Feature Extraction and De-Identification

#### 2.4.1. Feature Classification

Features extracted from the AX_T1+C DICOM images of both the cancer and non-cancer groups were classified as “Meta” (abnormal), indicating potential tumor metastasis, or “No_abnormalities” (normal), suggesting the absence of detectable abnormalities. These classified features were then stored in separate Excel files (Meta.xlsx and No_abnormalities.xlsx) to maintain organization and facilitate subsequent analyses. The labeling process was based on radiologist consensus review and corresponding clinical reports. The “Meta” group consisted of 100 cases with confirmed brain metastases, while the “No_abnormalities” group included 100 age- and gender-matched controls with no known cancer history.

A total of 1051 radiomic features were computed for each case, including original, Laplacian of Gaussian (LoG), and wavelet-derived features. Feature naming and trans-formation categories are documented in Appendix A.

These labeled feature sets formed the core input for downstream GPT-4o-based inference and were version-controlled to ensure reproducibility. CLAIM checklist Items 16 and 21 (see Appendix A) are addressed by this structured classification and data segregation process.

#### 2.4.2. Data Merging and De-Identification

Following the initial classification and storage, the two Excel files (Meta.xlsx and No_abnormalities.xlsx) were merged into a single, comprehensive dataset named No_abnormalities & Brain_Meta (De-identification).xlsx. This merged dataset contained both abnormal (Meta) and normal (No_abnormalities) samples, providing a unified data source for model training and evaluation. Prior to merging, all files were verified for format consistency, and feature columns were harmonized across the two classes to ensure compatibility. To ensure patient privacy and comply with data protection regulations, patient IDs and other sensitive information were removed from this dataset. Only the classification labels (Meta or No_abnormalities) were retained to preserve the essential information for model training and analysis. The final dataset included 200 subjects with 1051 radiomic features per case and was structured in tabular format for compatibility with both GPT-4o-prompting pipelines and external model evaluation.

The full data processing workflow is visualized in Appendix A, and this protocol adheres to the transparency and safety criteria described in CLAIM 2024 (see Appendix A, Items 18, 21, and 43).

### 2.5. Creating the Brain ARAMP

#### 2.5.1. Reference Integration

To enhance MRI interpretation, a retrieval-augmented generation (RAG) system was implemented.

This framework enabled GPT-4o to dynamically access five authoritative medical references [17,18,19,20,21], providing domain-specific knowledge to support structured lesion analysis and clinical reasoning. The RAG implementation employed a version-controlled, keyword-indexed knowledge base constructed from core neuroradiology textbooks (e.g., Osborn’s Brain, ESNR Textbook). During inference, lesion descriptors such as “ring-enhancing lesion with peritumoral edema” triggered semantic-based retrieval of relevant literature. The update frequency of this knowledge base aligned with published guideline revisions. The literature base was version-controlled and aligned with updated clinical guidelines.

#### 2.5.2. Dataset Validation

A structured dataset (No_abnormalities & Brain_Meta.xlsx) containing the extracted imaging features corresponding classifications was used for validation. The dataset included a “TYPE” column indicating feature categories (e.g., original_shape_VoxelVolume), allowing for easy identification and retrieval of specific feature categories. The “No_abnormalities” and “Brain_Meta” feature values enabled direct comparison and analysis of the feature distributions between the two groups. Radiomic features were extracted using PyRadiomics v3.0, with key settings including voxel resampling to [1, 1, 1] mm (B-spline interpolation), intensity normalization (scale: 100, shift: 1000), discretization bin width = 25, LoG filtering (σ = 3 and 5 mm), and Haar wavelet decomposition (level 1). Feature classes included first-order statistics, shape descriptors, and texture-based features. Full extraction parameters are provided in Appendix A.

#### 2.5.3. Image Data and Workflow

Composite images, each containing up to 10 sequentially arranged MR images, were used as the input for the GPT model. These composite images provided a comprehensive representation of the patient’s brain anatomy and pathology, supporting structured diagnostic reasoning by the model. A structured workflow, consisting of four distinct steps, was implemented to guide the brain MRI process by the GPT model. This process was applied to a focused evaluation cohort consisting of 100 cancer patients with confirmed brain metastases, as determined by radiologist consensus and original MRI reports. The inclusion of only metastasis-positive cases was intentional to assess reasoning improvement rather than classification accuracy; thus, specificity and overall accuracy were not computed.

Step 1: Image Definition and Assessment: this initial step involved defining the imaging type (MRI), specifying the organ being scanned (brain), and setting appropriate MRI scan parameters (e.g., T1-weighted, T2-weighted; axial, sagittal, coronal views; pre-contrast and post-contrast) based on the clinical context and diagnostic objectives.Step 2: Lesion Analysis and Differential Diagnosis: in this step, the model analyzed any detected lesions, characterizing their size, intensity, and enhancement patterns. It then developed a differential diagnosis based on these lesion characteristics, dynamically retrieving and referencing relevant diagnostic criteria from the integrated medical literature [17,18,19,20,21]. Lesion descriptors served as semantic triggers within the RAG system to retrieve guideline-based diagnostic criteria, enabling precise matching with literature-defined imaging patterns.Step 3: Treatment Strategy for Brain Metastases: if brain metastases were suspected or confirmed, the model proceeded to formulate a potential treatment plan based on the imaging findings and diagnosis. This involved matching the imaging features with established treatment criteria and dynamically integrating relevant treatment recommendations from the medical literature [17,18,19,20,21]. Literature retrieval was tailored through prompt-linked indexing aligned with oncology treatment standards, allowing context-specific adaptation.Step 4: Conclusion and Follow-Up: in the final step, the model summarized the diagnostic findings, proposed a recommended treatment plan (if applicable), and defined appropriate follow-up requirements, tailoring imaging intervals and plans using dynamic references to the medical literature [17,18,19,20,21]. Each output included traceable reasoning and literature-cited justifications to enhance interpretability and transparency.

This inference protocol reflects ARAMP’s intended role as a decision-support tool applied after radiological abnormalities are identified, focusing on interpretive accuracy rather than primary detection.

### 2.6. Inquiry Procedure and Evaluation

To evaluate the performance of the developed Brain ARAMP, a structured inquiry procedure was conducted. The evaluation focused on comparing ARAMP’s accuracy and consistency to interpretations by expert radiologists, alongside performance benchmarks from a foundation LLM (Gemini) and relevant baseline models. This evaluation was conducted exclusively on a cohort of 100 patients with imaging-confirmed brain metastases, as verified through radiologist consensus review. Control cases were not included in this phase, as the analysis emphasized interpretive refinement rather than classification. This procedure involved two phases of inquiries:GPT-4o Inquiry: An initial inquiry was made in GPT-4o, a large language model, to generate a baseline analysis report for each set of “Meta” images. This baseline assessment served as a reference point for comparison with the subsequent RAG-enhanced analysis.Brain ARAMP Inquiries: Following the GPT-4o inquiry, a series of four inquiries was performed in the Brain ARAMP. These inquiries were conducted at different stages of the model’s training process:
(1)After Training: The first inquiry assessed the model’s performance immediately after the initial training phase.(2)Post-Training 1, 2, and 3: The subsequent inquiries evaluated the model’s performance after three rounds of iterative post-training refinement, allowing for assessment of the model’s learning and improvement over time.

During each inquiry, a standardized prompt was applied to consistent analysis across all cases. The prompt explicitly instructed the model to execute the full diagnostic workflow comprehensively and step by step, spanning Step 1 (Image Definition) to Step 4 (Follow-Up Planning). To ensure transparency and facilitate evaluation, the prompt also required the model to embed all relevant programming code, intermediate results, and metadata validations in its output. Additionally, the model was instructed to provide clear reasoning for every decision, referencing retrieved dataset values, authoritative literature, or clinical guidelines [17,18,19,20,21].

It is important to clarify that ARAMP was evaluated as a reasoning-based support tool—not a detection model. Its role is to assist in post-detection interpretation of abnormal MR images, particularly in patients with known or suspected metastatic disease. This evaluation strategy, focused solely on metastasis-confirmed cases, allowed for precise assessment of interpretive enhancement while avoiding misclassification bias from unlabeled negatives. Expert evaluation of the model’s diagnostic performance was conducted using several metrics:Pre-Trained Inference Difference: Assesses the discrepancy between initial model outputs (prior to structured prompting) and expert-confirmed diagnoses, illustrating baseline diagnostic variance.Inter-Inference Agreement: Measures consistency among different post-training model outputs across three prompt iterations, indicating the stability and reproducibility of ARAMP’s diagnostic reasoning.Post-Trained Inference Difference: Evaluates deviation between refined model outputs (after full RAG-enhanced prompting) and expert diagnoses, serving as a proxy for overall interpretive accuracy.

All metrics were scored on a 0–100% scale, where 0% indicates total discordance or incorrect reasoning, and 100% denotes complete agreement with expert assessment.

### 2.7. Statistical Analysis

Descriptive statistics (e.g., mean ± standard deviation, median [range], frequencies, and percentages) were used to summarize the demographic and clinical characteristics of the patient subgroups, as presented in Table 1. All statistical analyses were performed using IBM SPSS Statistics for Windows, version 26.0 (IBM Corp., Armonk, NY, USA). Given the focus on interpretive improvement rather than statistical significance testing, no inferential statistical tests (e.g., Wilcoxon, *t*-test) were applied. The pre- and post-training comparisons were based on aggregated descriptive measures and scoring trends across expert raters, as shown in Appendix A. Inter-rater agreement was assessed descriptively using intraclass correlation coefficients (ICC), as summarized in Table 2.

## 3. Result

### 3.1. Patient Characteristics

A total of 2447 patients (cancer group) with confirmed cancer diagnoses who underwent post-contrast axial T1-weighted brain MRI (AX_T1+C) examinations between 2010 and 2022 were identified from the institutional PACS system. The median age was 64 years (range, 21–95 years); 1323 of the patients (54.1%) were male.

From this cohort, a random subset of 100 patients was selected for detailed analysis. These patients had documented brain metastases (Mets) in their clinical MRI reports, further confirmed by expert consensus review (100% agreement).

In addition, 100 non-cancer individuals were selected as a control group, matched to the cancer group by age and gender distribution.

Although not included in model performance comparison, this control group underwent radiomic feature extraction to support construction of the RAG-based reference matrix for differential prompting.

Together, the 200 cases (100 cancer, 100 control) comprised the full dataset used for radiomic analysis and feature-informed GPT inference.

The demographic characteristics of the selected cancer subsample are presented in Table 1.

Radiologist-based evaluations demonstrated improved clarity and diagnostic consistency in ARAMP-generated reports, with detailed results presented in Appendix A.

### 3.2. DICOM Feature Extraction and Image Conversion

A total of 1051 quantitative imaging features were extracted from the AX_T1+C DICOM images using the PyRadiomics library. These features included first-order statistical features (e.g., mean, skewness, kurtosis), shape-based features (e.g., volume, sphericity), as well as Laplacian of Gaussian (LoG) and wavelet-transformed features.

All extraction parameters, including voxel resampling, normalization, bin width, and enabled feature classes, are detailed in Appendix A.

The extracted features were not used to train a conventional deep learning model, but rather served as structured input for GPT-4o in a retrieval-augmented diagnostic reasoning framework.

Following intensity normalization, axial DICOM images were converted into PNG format. Composite images were then constructed, with each composite comprising up to 10 sequentially arranged MRI slices, representing axial anatomical context per case. These composite PNGs were paired with their corresponding radiomic features and served as multimodal input for GPT-based zero-shot inference.

### 3.3. GPT Model Performance

This study evaluated the performance of the Brain ARAMP in identifying and reporting brain metastases among 100 cancer patients. Two key measures, derived from independent ratings by three board-certified radiologists and supplemented by Gemini benchmark outputs, are summarized in Table 2 and Table 3:Post-Trained Inference Difference: The model’s post-trained inference difference was 67.45%, reflecting a substantial improvement in interpretive quality following implementation of the ARAMP prompting framework. In this context, “training” refers to a structured, multi-round prompting process—incorporating radiomic features, composite images, and literature-assisted reasoning via Retrieval-Augmented Generation (RAG)—rather than any form of parameter fine-tuning or supervised model updates.Inter-Inference Agreement: The average inter-inference agreement between the three expert radiologists was 30.20%, reflecting modest exact-match consistency in their scoring of the model’s diagnostic outputs. To formally assess inter-rater reliability, the intraclass correlation coefficient (ICC) was computed, yielding a statistically significant result (ICC = 0.192, *p* = 0.01).

This agreement suggests that the model’s post-training reports were meaningfully distinct and consistently perceived as superior, supporting the reliability of the ARAMP-enhanced GPT workflow.

**Table 3 bioengineering-12-00698-t003:** Evaluator-level post-trained inference similarity and exact-match agreement.

Evaluator	Post-Trained Inference Difference (Mean ± SD)	Inter-Inference Agreement (Mean ± SD)
Dr. A	56.20 ± 25.54	22.80 ± 8.54
Dr. B	73.20 ± 17.57	40.40 ± 13.33
Dr. C	75.60 ± 18.77	35.40 ± 11.67
Gemini	68.00 ± 22.74	22.00 ± 6.03
Average of Drs. A–C	68.33 ± 20.63	32.87 ± 11.18

Note: All values are presented as percentages (mean ± standard deviation, SD).

In addition, comparisons with the expert-reviewed gold standard revealed that the model achieved a sensitivity of 0.84 before training and 0.98 after training, reflecting a marked improvement in correctly identifying positive cases (see Table 4). As previously defined, “training” here refers to the structured prompting strategy incorporating radiomic features, composite MR images, and retrieval-augmented literature support, without any parameter fine-tuning.

Specificity and overall accuracy were not reported, due to the absence of negative cases in the dataset; in this context, accuracy is mathematically equivalent to sensitivity, and specificity could not be meaningfully calculated.

Furthermore, integration of the RAG framework significantly improved the model’s reliability in prioritizing brain metastases as the top-ranked differential diagnosis, as detailed in Table 5.

#### 3.3.1. Mention Type Definitions

Listed as First: Brain metastasis was explicitly ranked as the first differential diagnosis.Listed but Not First: Brain metastasis was included in the differential diagnoses, but not was not listed first.Not Mentioned: Brain metastasis was not included in the differential diagnosis at all.

#### 3.3.2. Post-Trained Inference Difference

This analysis aimed to assess whether the implementation of structured prompting strategies—as previously defined in the ARAMP workflow—led to measurable improvements in reporting quality. For each MRI examination, the GPT model generated one baseline report (pre-training) and three post-training reports, yielding four distinct outputs per MRI examination. These reports were independently evaluated by three board-certified neuroradiologists, each scoring them on predefined clinical criteria. ICC was calculated to assess the extent to which the ratings reflected meaningful differences among the reports. The results indicated low but statistically significant agreement between raters: ICC (single measures) = 0.286, 95% CI [0.161, 0.415], *p* = 0.000; ICC (average measures) = 0.546, 95% CI [0.366, 0.681], *p* = 0.000. While overall rater agreement was limited, the statistically significant ICC values suggest that the post-training reports were, to some extent, distinguishable from pre-training reports in terms of quality (Table 6).

#### 3.3.3. Inter-Inference Agreement

To examine the stability of model outputs given identical input, three post-training reports generated from the same image were evaluated by the two radiologists. ICC values indicated statistically significant inter-rater agreement (Table 6): ICC (single measures) = 0.192, 95% CI [0.071, 0.324], *p* = 0.01; ICC (average measures) = 0.417, 95% CI [0.186, 0.590], *p* = 0.01. These results reflect statistically significant consistency between the raters in evaluating the similarity of the model outputs, indicating a fair-to-moderate level of stability in the model’s output based on physician ratings.

To qualitatively evaluate diagnostic improvement, structured annotations were obtained from three board-certified neuroradiologists—referred to as Reviewer A, Reviewer B, and Reviewer C––across 100 MRI cases (MRI1–MRI100). Each reviewer independently assessed the AI-generated diagnostic reports, focusing on correctness, granularity, and lesion-level consistency.

Reviewer A provided only three annotations. All were brief statements indicating a general preference for the pre-trained model (e.g., “pre-trained better”) and did not include specific lesion-level descriptions or model critique. His feedback served as high-level impressions/indications, rather than detailed evaluations.

Reviewer B offered the most comprehensive and detailed feedback. Across the dataset, 82 positive and 43 negative annotations were provided. Among them, 34 cases received both positive and negative remarks, 48 were annotated only with positive comments, and 9 were marked only with negative critiques. The remaining nine cases did not receive any annotations from Reviewer B. Positive feedback highlighted improvements in lesion localization, morphological detail, and accurate classification of lesion multiplicity. Examples included enhanced recognition of supratentorial/infratentorial spread, vasogenic edema, and ring-enhancing features. Negative comments focused on hallucinated findings, lateralization errors (e.g., “left diagnosed as right”), and missed lesions in key anatomical regions such as the cerebellum or skull.

Reviewer C contributed 21 annotations. These comments provided comparative judgments of pre- versus post-trained outputs and included lesion-level observations, such as “No lesion. Post-train is better,” “tiny lesion only identified on post-train,” or “Any certain lesion?”. In many cases, Reviewer C indicated whether a lesion was properly identified, missed, or ambiguously described. Several annotations offered insight into subtle or uncertain imaging findings, especially in borderline or very small lesions. His feedback provided contextual interpretation of model performance in diagnostically challenging scenarios.

This multi-reviewer annotation strategy not only ensured clinically grounded validation of AI-generated outputs, but also enabled a nuanced understanding of model strengths and limitations from diverse expert perspectives. The combined qualitative and quantitative evaluations supported a robust diagnostic performance assessment and informed iterative model refinement.

### 3.4. Illustrative Case

An example composite PNG was generated from 10 slices of AX_T1+C MR images from a patient with imaging diagnosis of brain metastases. A composite PNG with 10 slices of MR images is uploaded in every iteration of training due to capacity limitation. Depending on the slice thickness of the MR protocol, a patient’s images may need 6 to 18 iterations to complete the training (Figure 1).

## 4. Discussion

This study investigated the potential of a novel imaging framework, ARAMP (Adaptive RAG Assistant MRI Platform), to assist in detecting brain metastases in cancer patients undergoing brain MRI examinations. Our findings demonstrate the feasibility of extracting quantitative MRI data. By combining GPT-4o, structured radiomic features, and a retrieval-augmented generation (RAG) mechanism, ARAMP demonstrated promising diagnostic performance with both quantitative and qualitative validation.

The moderate inter-inference agreement observed suggests that the model’s diagnostic output is reasonably consistent and reproducible. Although a 30.20% exact-match agreement may initially appear low—suggesting that roughly 70% of rater responses were not perfectly aligned—it is important to interpret this figure in context. As clarified in the revised captions for Table 2 and Table 3, these values represent strict exact match rates (mean ± SD) across three independent raters, making this a deliberately conservative agreement measure. In tasks involving narrative diagnostic reasoning from AI systems, even subtle wording differences can lead to a mismatch, despite underlying clinical consensus. Similarly, the inter-round agreement scores reported in Table 3 (mean 32.87% ± 11.18%) reflect the average pairwise exact-match rate among repeated evaluations of the same post-trained outputs. These scores further underscore the challenge of achieving identical phrasing across multiple raters or evaluation rounds, especially when judging free-text outputs derived from LLMs. Importantly, this metric captures rater consistency in evaluating model outputs rather than model variability itself.

Prior studies of free-text or natural language outputs in clinical AI applications have reported similarly modest inter-rater agreement (kappa = 0.77–0.91 for human evaluators, 0.69–0.87 for human–machine agreement) due to inherent interpretive variability in clinical concept extraction and natural language processing tasks [22]. To supplement exact-match metrics, we also computed the intraclass correlation coefficient (ICC = 0.192, *p* = 0.01; see Table 6), which showed a statistically significant, though modest, level of consistency beyond chance. Together, these results suggest that, while exact-match agreement was limited, the AI-generated inferences were sufficiently reproducible to support their clinical utility.

Moreover, the substantial improvement in diagnostic accuracy following the training interventions (as shown by the increase from pre-trained to post-trained performance of 31.2 to 69.8%) underscores the effectiveness of ARAMP in enhancing the model’s learning and performance.

The integration of authoritative medical references through the RAG system allows the model to access and utilize a wealth of domain-specific knowledge, offering interpretive capabilities beyond those of rule-based or standalone models lacking external knowledge grounding. This dynamic integration of literature enables the model to adapt to new information and refine its diagnostic capabilities over time. Compare these findings in detail with specific results from studies like Pristoupil et al. [15] and Sau et al. [16] highlights how the ARAMP approach potentially addresses the limitations of LLMs in image-based tasks, as noted by Sau et al. Also compare with other AI/radiomics studies for brain metastasis detection [9,10], which discuss differences in methodology and performance.

We made some observations regarding model hallucination, and the adaptive RAG mechanism, combined with the structured workflow and reference grounding, appeared to mitigate this risk in our study.

While our results are encouraging, this study has limitations. First, the retrospective design introduces potential biases related to patient selection and data quality. Second, the sample size in the subgroup analysis (*n* = 100) may limit the generalizability of our findings regarding detailed performance metrics. The study was conducted at a single institution; the gold standard relied on internal MRI and radiology reports, which might have inherent limitations. Third, the model was tested exclusively on AX_T1+C sequences. This design choice reflects clinical consensus that contrast-enhanced T1-weighted imaging offers optimal sensitivity for brain metastasis detection, as supported by prior studies such as Grøvik et al. (2020), Pennig et al. (2022), and Zhang et al. (2022) [23,24,25]. While additional sequences like FLAIR or DWI may provide supplementary specificity in complex cases, AX_T1+C remains the most validated modality for lesion conspicuity and radiomic analysis. Recent studies, such as Machura et al. (2024) [26], have demonstrated that incorporating longitudinal multi-sequence MRI may improve detection consistency in deep learning ensembles. Although our current study did not evaluate FLAIR or DWI, we acknowledge their potential value, and the integration of multimodal imaging data will be a focus of future ARAMP iterations.

Because the evaluation cohort consisted solely of metastasis-positive cases confirmed by expert review, specificity and overall accuracy metrics were not calculated. Including unlabeled or unverified control scans could introduce verification bias or misclassification, thereby compromising analytic validity. Sensitivity, which measures true positive identification within known abnormal cases, was selected as the most appropriate metric to assess interpretive refinement. This approach aligns with ARAMP’s post-detection role, focusing on diagnostic reasoning rather than binary classification. Due to the absence of negative (failing) cases in the evaluation dataset, overall accuracy was not a meaningful metric and was not reported. Sensitivity was used instead to assess the model’s ability to correctly identify passing cases. Prospective studies are also needed, in the future, to evaluate the real-world clinical impact.

Despite these limitations, this study highlights the potential of LLM-based systems with adaptive RAG to assist radiologists in the interpretation of brain MRI images and the detection of brain metastases. Further research is warranted to explore the clinical utility of this technology in improving diagnostic accuracy and, subsequently, patient outcomes.

This study presents development and evaluation of ARAMP, a system integrating a large language model (GPT-4o) with a retrieval-augmented generation (RAG) framework and multiple domain-specific medical references. The system supports information retrieval and AI-assisted content generation. We assessed ARAMP’s efficacy in generating radiology reports for MRI studies, comparing its outputs before and after training by GPT-4o, using both human expert review and independent LLM-based inference.

Radiological workloads continue to increase amid a global shortage of trained radiologists. Recent advancements in LLMs and RAG architectures offer potential solutions to alleviate interpretive burden. This study investigates the clinical relevance and performance of ARAMP, an LLM-RAG integrated platform designed to assist radiologists in MRI interpretation. ARAMP leverages GPT-4o in conjunction with curated medical references [17,18,19,20,21] to perform AI-assisted MRI interpretation. Three board-certified radiologists with varying years of clinical experience independently evaluated the generated reports. An independent LLM (Gemini), which was not involved in the training process, was also used to assess inference performance. Key metrics included model interpretative score (pre- vs. post-training), inter-rater consistency, and parity between human and machine inference. Post-training, the performance of ARAMP improved significantly, with interpretative scores increasing from 59.90 to 67.45. The average inter-inference consistency among the three radiologists was 30.20 out of 100 (*p* = 0.01, Table 2), indicating a moderately high agreement level. Differences between pre- and post-training inference results were comparable between radiologists (average: 68.00/100.00) and Gemini (68.33/100.00), suggesting reliable model generalization and parity in independent assessment. These results provide empirical evidence that LLM-based models like ARAMP can achieve near-human performance in evaluating AI-generated content (Table 3). Moreover, Gemini’s independent assessment validates the generalizability of the post-trained ARAMP model across different inference platforms.

ARAMP demonstrates promising performance in AI-assisted MRI interpretation, supported by expert evaluation and independent model validation. Its integration into radiological practice could help address the increasing demand for imaging interpretation and improve clinical efficiency. The implementation of ARAMP in clinical settings may offer significant benefits by streamlining MRI interpretation workflows. As a previewing mechanism, it has the potential to reduce the diagnostic workload of radiologists—particularly important in an era marked by radiologist shortages worldwide.

## 5. Conclusions

Our proposed LLM model, Brain ARAMP, demonstrates potential to assist in the detection of brain metastases in cancer patients undergoing MRI examinations. The model demonstrated promising diagnostic accuracy and statistically significant inter-inference agreement (indicating moderate-to-high consistency) among expert radiologists. Future research should focus on validating these findings in larger, more diverse patient populations with brain tumors. The research scope can also prospectively extend to evaluating the model’s impact on clinical decision-making and patient outcomes.

## Figures and Tables

**Figure 1 bioengineering-12-00698-f001:**
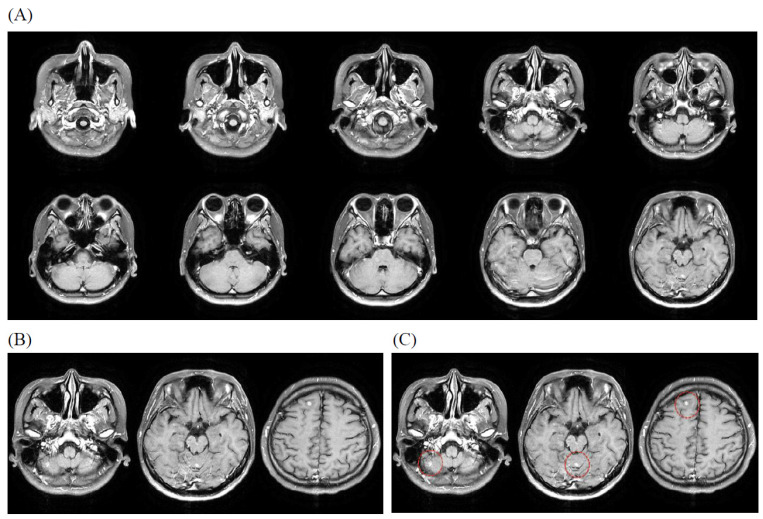
Illustrative case of brain metastases evaluated by the ARAMP system. (**A**) Composite PNG image showing 10 sequential post-contrast T1-weighted axial (AX_T1+C) MR slices. (**B**) Three representative AX_T1+C slices demonstrating multiple metastatic lesions (examples indicated by red circles in panel (**C**). Note the lesion in the right cerebellum showing significant surrounding vasogenic edema (red circle). (**C**) Magnified view of the slices in (**B**), with metastatic lesions clearly marked by red circles. These lesions typically appear as enhancing nodules on post-contrast T1-weighted images.

**Table 1 bioengineering-12-00698-t001:** Demographic and clinical characteristics of the 100-patient subsample used for metastasis evaluation.

Characteristics	Value
Gender (male/female)	52/48
Age (male/female)	
20–30	1/1
30–40	2/2
40–50	5/6
50–60	12/12
60–70	17/14
70–80	12/9
80+	3/4
Clinical diagnosis	
Biliary or pancreatic cancer	1
Breast cancer	11
Cervical cancer	1
Colorectal cancer	3
Esophageal carcinoma	3
Gastric cancer	1
Head and neck cancer (excluding NPC_brain_spine_thyroid cancer)	2
Liver cancer	1
Lung cancer	65
Melanoma	1
Non-Hodgkin’s disease	3
Other cancers	3
Prostate cancer	2
Secondary malignant neoplasm	1
Upper lower respiratory tract cancer (excluding lung cancer)	1
Urologic cancer (excluding renal_bladder_prostate cancer)	1

Note. Age distribution is shown as male/female counts.

**Table 2 bioengineering-12-00698-t002:** Summary of pre- and post-trained inference performance and inter-rater agreement.

Measure	Pre-Trained	Post-Trained	Inter-Inference Agreement
Mean ± SD	59.90 ± 17.99	67.45 ± 21.60	30.20 ± 12.98
Sum of Converted Scores	11,980	26,980	12,080
Range (Min–Max)	20–100	20–100	20–100

Note: Scores were converted from a 5-point scale (3 raters × 100 cases = 300 ratings per group) into percentage categories (20–100%). Values represent the average across all evaluators and cases.

**Table 4 bioengineering-12-00698-t004:** Comparison of instance level model performance before and after ARAMP training.

Metric	Before	Post
True Positive (TP)	84	98
False Negative (FN)	16	2
True Negative (TN)	0	0
False Positive (FP)	0	0
Sensitivity	0.84	0.98
Specificity	NA	NA
Precision	84%	98%

Note: Sensitivity improved after training. Specificity could not be calculated, due to the absence of negative cases in the dataset.

**Table 5 bioengineering-12-00698-t005:** Distribution of brain metastasis mentions across model outputs.

Mention Type	Before	Post 1	Post 2	Post 3
Listed as First	78	95	95	95
Listed but Not First	6	3	3	3
Not Mentioned	16	2	2	2

Note: Before refers to the initial model output without retrieval augmentation. Post 1–Post 3 are reports generated after applying RAG (retrieval-augmented generation) prompting.

**Table 6 bioengineering-12-00698-t006:** Intraclass correlation coefficient (ICC) for physician ratings.

Analysis Context	ICC Type	Intraclass Correlation ^b^	95% CI (Lower–Upper)	F (df1, df2)	*p*-Value
Post-trained inference differences	Single measures	0.286 ^a^	0.161~0.415	2.200 (99, 198)	0.00
	Average measures	0.546 ^c^	0.366~0.681	2.200 (99, 198)	0.00
Inter-inference agreement of post-trained model	Single measures	0.192 ^a^	0.071~0.324	1.714 (99, 198)	0.01
	Average measures	0.417 ^c^	0.186~0.590	1.714 (99, 198)	0.01

Note: Two-way mixed effects model where people effects are random and measures effects are fixed. ^a^ The estimator is the same, whether the interaction effect is present or not. ^b^ Type C intraclass correlation coefficients using a consistency definition. Between-measure variance is excluded from the denominator variance. ^c^ This estimate is computed assuming the interaction effect is absent, because it is not estimable otherwise.

## Data Availability

The datasets generated and/or analyzed during the current study are not publicly available, due to patient privacy regulations, but are available from the corresponding author on reasonable request and with institutional permission.

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
