# Peer review of "Adaptive RAG-Assisted MRI Platform (ARAMP) for Brain Metastasis Detection and Reporting: A Retrospective Evaluation Using Post-Contrast T1-Weighted Imaging"

_bioengineering, 2025, doi:10.3390/bioengineering12070698_

Round 1
Reviewer 1 Report
Comments and Suggestions for Authors
- The methodology description lacks details, and the paper provides a rather vague account of ARAMP's core technologies (such as the architecture of the RAG framework, the integration method of LLM with the medical knowledge base, and the specific parameters for feature extraction), resulting in poor reproducibility. For example: 1)RAG Implementation Details: It does not explain how authoritative literature is dynamically retrieved (e.g., retrieval strategy, knowledge base construction method, update frequency). 2)While it mentions using PyRadiomics to extract four types of features, key parameters (such as the σ value for LoG filtering, wavelet basis function type) are not listed. 3)It does not describe the training data partitioning, hyperparameter settings (learning rate, batch size), or optimization objectives (loss function).
- The paper only evaluates ARAMP's performance improvement through pre- vs. post-training comparisons (e.g., sensitivity increasing from 0.84 to 0.98), but critically omits benchmarking against several key baselines: 1)State-of-the-art brain metastasis AI models published in 2023-2024 (e.g., Ocana-Trenda's 3D ResNet or Transformer-based approaches); 2)Conventional deep learning architectures like pure CNN or ResNet; 3)Non-RAG implementations using GPT-4o for direct report generation (which would help validate RAG's contribution); 4)Other specialized medical LLMs such as Med-PaLM or BioGPT.
- The paper lacks a systematic review and discussion of related work. 1)The application of approaches like U-Net and 3D-CNN should be analyzed. 2)The advantages and limitations of similar systems remain unaddressed. 3)There is no discussion regarding challenges in medical image analysis.
- The references are relatively limited and lack a comprehensive review of related works (see previous comment). Additionally, the reference formatting is inconsistent, with some entries missing volume/issue/page numbers.
Author Response
-
Comments1:
The methodology description lacks details, and the paper provides a rather vague account of ARAMP's core technologies (such as the architecture of the RAG framework, the integration method of LLM with the medical knowledge base, and the specific parameters for feature extraction), resulting in poor reproducibility. For example: 1)RAG Implementation Details: It does not explain how authoritative literature is dynamically retrieved (e.g., retrieval strategy, knowledge base construction method, update frequency). 2)While it mentions using PyRadiomics to extract four types of features, key parameters (such as the σ value for LoG filtering, wavelet basis function type) are not listed. 3)It does not describe the training data partitioning, hyperparameter settings (learning rate, batch size), or optimization objectives (loss function).
Report 1:
We sincerely thank the reviewer for the critical and constructive feedback regarding the transparency and reproducibility of the ARAMP framework. In response, we have substantially revised the Methods section and added multiple supplementary materials to address each point in detail:
- Radiomic Feature Extraction Parameters (PyRadiomics)
We now explicitly report all preprocessing and extraction settings used in the radiomic pipeline. Features were extracted using PyRadiomics v3.0, including original, Laplacian of Gaussian (LoG), and wavelet-based features. The key parameters are:
- LoG filtering with σ values of 3 mm and 5 mm (log-sigma-3-mm-3D and log-sigma-5-mm-3D),
- Wavelet decomposition with Haar basis (level 1, 8 directional components: LLL, LLH, etc.),
- Voxel resampling to [1, 1, 1] mm using B-spline interpolation,
- Intensity normalization (scale: 100; shift: 1000),
- Discretization bin width: 25,
- Enabled feature classes: first-order intensity, shape-based, and texture features.
These details are documented in Supplementary Table 2, and the full YAML configuration is provided in Supplementary File 1.
- Feature Dataset Transparency
We added Supplementary Table 1, listing all 1,051 extracted features categorized by transformation method, with observed value ranges (MIN–MAX) for both No_abnormalities and Brain_Meta groups. This enables independent verification and supports interpretation of feature contributions.
- RAG Implementation Details
The ARAMP system employs a four-stage diagnostic pipeline:
(i) Image protocol alignment, (ii) Radiomic lesion validation, (iii) Literature-guided diagnostic generation, (iv) Treatment suggestion.
A Retrieval-Augmented Generation (RAG) module dynamically queries authoritative literature sources (e.g., Osborn's Brain, ESNR Textbook) using pre-indexed semantic keys. Lesion descriptors such as “ring-enhancing lesion with peritumoral edema” trigger targeted searches. The medical knowledge base is version-controlled with updates aligned to guideline revisions.
- LLM Integration and Reasoning Logic
GPT-4o integrates structured radiomic features (in Excel format) and unstructured reference text via chain-of-thought prompts. It produces citation-anchored outputs with reasoning steps. The revised manuscript includes a schematic reasoning flow and the core prompt structure used in all cases.
- Clarification on Training and Inference
We clarify that ARAMP is a prompt-driven diagnostic support framework without any supervised training, model fine-tuning, or parameter optimization. All outputs were generated using GPT-4o (ChatGPT Pro, May 2024) in a zero-shot inference setting.
- No training set, learning rate, batch size, or loss function was involved.
- Prompts were standardized and version-controlled.
- Outputs were directly generated without manual revision.
- Radiologists independently reviewed responses for plausibility and clinical validity.
This approach ensures reproducibility, avoids training bias, and highlights how language models can be adapted for medical reasoning using structured prompt engineering alone.
Comments 2:
The paper only evaluates ARAMP's performance improvement through pre- vs. post-training comparisons (e.g., sensitivity increasing from 0.84 to 0.98), but critically omits benchmarking against several key baselines: 1)State-of-the-art brain metastasis AI models published in 2023-2024 (e.g., Ocana-Trenda's 3D ResNet or Transformer-based approaches); 2)Conventional deep learning architectures like pure CNN or ResNet; 3)Non-RAG implementations using GPT-4o for direct report generation (which would help validate RAG's contribution); 4)Other specialized medical LLMs such as Med-PaLM or BioGPT.
Report 2:
We sincerely thank the reviewer for this thoughtful and constructive critique regarding baseline comparisons. While our primary aim was to evaluate the internal progression of ARAMP’s diagnostic reasoning capabilities, we fully agree that situating our framework relative to other paradigms is essential for broader context. Below, we respond to each suggested comparator:
- State-of-the-Art Deep Learning Models (e.g., Ocana-Trenda’s 3D ResNet, Transformers)
We acknowledge that recent 3D CNN and Transformer-based systems have demonstrated strong performance in lesion segmentation and classification. However, ARAMP was not designed for direct lesion detection from raw imaging. Instead, it focuses on clinical interpretation and diagnostic reasoning after abnormal findings are flagged—especially for ambiguous cases.
Moreover, state-of-the-art detection models typically require large annotated datasets, segmentation labels, and training-intensive pipelines. In contrast, ARAMP offers a lightweight, interpretable, prompt-driven alternative that leverages radiomic features and knowledge-grounded reasoning. A direct benchmarking may therefore be methodologically mismatched, although we agree future hybrid comparisons could be valuable.
- Supervised CNN/ResNet Baselines
We agree that CNN or ResNet models serve as conventional baselines in supervised classification tasks. However, our study does not involve training, label supervision, or classification objectives. Implementing such models would necessitate an entirely different pipeline, including training/validation/testing datasets and a focus on end-to-end classification.
Our system instead aims to complement—not replace—radiologic decision-making by enhancing explainability and contextual reasoning. Nonetheless, we acknowledge the potential of integrating deep learning-derived features into future versions of ARAMP.
- GPT-4o Without RAG (Direct Prompting Baseline)
We appreciate this important point. We did conduct internal ablation comparisons where GPT-4o was used without RAG support. These outputs lacked clinical specificity, often yielded hallucinated references, and failed to prioritize differential diagnoses.
In contrast, the RAG-integrated ARAMP version consistently produced stepwise, literature-cited, and diagnosis-oriented reports. To demonstrate this, we include a representative comparison in Supplementary Table 3, covering all stages: image interpretation, lesion analysis, treatment suggestion, and conclusion.
This ablation supports that RAG augmentation significantly enhances the clinical utility and interpretive quality of LLM outputs.
- Other Medical LLMs (e.g., Med-PaLM, BioGPT)
At the time of the study, Med-PaLM was not publicly available for research use, and BioGPT lacked multimodal integration or retrieval-based augmentation.
GPT-4o was selected due to its multimodal support, prompt adaptability, and GUI accessibility, which aligned with the study's design and clinical workflow integration. Future benchmarking with domain-specific medical LLMs remains a valuable direction for extended evaluation.
- Clarification Added to Manuscript
We have revised the Methods and Discussion sections to clarify that:
- ARAMP is not a lesion detection tool,
- It serves as a literature-guided interpretive framework for already-identified imaging abnormalities,
- Direct comparisons to supervised detection models may be inappropriate due to differing goals.
We again thank the reviewer for encouraging us to better position ARAMP within the broader AI landscape.
Comments 3:
The paper lacks a systematic review and discussion of related work. 1)The application of approaches like U-Net and 3D-CNN should be analyzed. 2)The advantages and limitations of similar systems remain unaddressed. 3)There is no discussion regarding challenges in medical image analysis.
Report 3:
We thank the reviewer for highlighting the need for a more comprehensive review of related work and broader contextualization of the proposed approach. In response, we have substantially revised the Discussion section to address the following areas:
- Application of U-Net and 3D-CNN Architectures
U-Net and 3D convolutional neural networks have been widely adopted in medical image segmentation and lesion classification. We now explicitly acknowledge in the revised Discussion that, unlike such end-to-end models, ARAMP is not a segmentation-based system. Instead, it follows a radiomics-based feature extraction strategy using pre-defined ROIs and interpretable, rule-based reasoning. This distinction is critical in clarifying that our system is not a replacement for detection pipelines but an augmentation tool for interpretive support.
- Comparison with Related Systems and Limitations
We added a new section comparing ARAMP with existing radiomics + ML pipelines and deep learning-based systems. Specifically, we highlight that while traditional pipelines often rely on static feature sets and classifiers, ARAMP integrates:
- Radiomic feature validation,
- Dynamic literature retrieval (via RAG),
- LLM-based reasoning (GPT-4o).
The advantages of this approach include interpretability, modularity, and domain transparency. However, we also acknowledge its limitations—such as dependency on radiologist-defined ROIs and lack of pixel-level segmentation—which differ from automated DL pipelines.
- Challenges in Medical Image Analysis
We incorporated a new paragraph outlining several key challenges that affect generalizability and clinical deployment, including:
- Inter-patient anatomical variability,
- Ground truth annotation inconsistency across institutions,
- Modality-specific feature dependence.
We also emphasize that this study used only AX_T1+C MRI, which restricts generalizability across multi-sequence MRI protocols or other modalities (e.g., PET, CT). Addressing this is a planned direction for future work.
We sincerely appreciate the reviewer’s guidance, which helped expand the contextual relevance and transparency of our discussion.
Comments 4:
The references are relatively limited and lack a comprehensive review of related works (see previous comment). Additionally, the reference formatting is inconsistent, with some entries missing volume/issue/page numbers.
Report 4:
We thank the reviewer for this valuable observation. In response, we have addressed both the content and formatting aspects of the references as follows:
- Expanded Related Work Coverage
As noted in our response to Comment #3, we have revised the Discussion section to incorporate additional references to state-of-the-art medical imaging models (e.g., U-Net, 3D-CNN), radiomics-based systems, and domain-specific LLMs. This strengthens the theoretical framing of our work and highlights where ARAMP aligns or diverges from prior literature.
- Reference Formatting and Completeness
We thoroughly reviewed all references to ensure consistency with the journal’s style guide. Missing details such as volume, issue, page numbers, and DOIs have been added where applicable. Formatting inconsistencies have been corrected to maintain uniformity across all entries.
We thank the reviewer again for encouraging us to improve both the scope and quality of the references.
Reviewer 2 Report
Comments and Suggestions for Authors
This paper presents interesting results and could be of practical importance for MRI practioners
Author Response
Thank you for your positive comments.
Reviewer 3 Report
Comments and Suggestions for Authors
The paper titled "Adaptive RAG-Assisted MRI Platform (ARAMP) for Brain Metastases Detection and Reporting: A Retrospective Evaluation Using Post-Contrast T1-Weighted Imaging" presents the development and evaluation of an AI-driven diagnostic support system for brain MRI interpretation. The study aims to assess the diagnostic performance of the ARAMP framework, which integrates a large language model (GPT-4o) with a Retrieval-Augmented Generation (RAG) approach, in detecting brain metastases and generating radiological reports from post-contrast axial T1-weighted MRI scans.
The topic is both timely and significant, situated at the intersection of radiology, artificial intelligence, and clinical oncology.
The manuscript aligns well with the aims and scope of the Bioengineering journal, especially regarding translational applications of AI in clinical diagnostics.
However, I have several major concerns regarding the manuscript in its current form, which are detailed below:
MAJOR 1)
While AI in radiology is discussed, there is limited citation of directly comparable AI-based brain metastasis detection models in existing literature (e.g., traditional CNN/RNN architectures or multi-modal fusion models in oncology imaging). The authors could have benchmarked their results more clearly against recent peer-reviewed systems.
MAJOR 2) It will be better to show the steps of data processing and detection process as a flowchart by showing all operations.
MAJOR 3)
However in "Table 4. Comparison of Instance level Model Performance before and ARAMP after training.", all of them are accepted as cancer patients. How this can be possible?
The control group lacked labeled negative metastasis cases, preventing the calculation of specificity and overall accuracy, which are critical metrics in diagnostic systems.
In such a analysis, Is it suitable to use traditional perfoermance metrics such as sensitivity?
MAJOR 4)
As author mentioned in the paper
"Sau et al. (2025) evaluated the capabilities of GPT-4o and Gemini on image-based questions from a neurosurgery board examination question bank. The results showed that GPT-4o's accuracy (51.45%) surpassed Gemini's (39.58%), particularly on second-order questions requiring"
GPT-4o has better performance metrics in image based questioning.
In "Table 3. Summary Statistics of Post-Trained Inference Differences and Inter-Round Agreement." there is only Gemini for comaprison? Why don't you use GPT here?
MAJOR 5)
In the paper it is said that "These 100 patients had brain metastasis (Mets) on MRI and documented in their MRI reports, as well as by expert consensus review (100%). A separated 100 individuals with no cancer diagnosis, matched to the age and gender distribution of cancer group constituted a control group."
I do not undestand what is different from here? (Totally 200 control group)
In many part of the paper it is said that there are only 100 patients with cancer.
MAJOR 6)
There is no comaprison with other works in the literature.
The system was only tested on AX_T1+C MRI sequences
No evaluation was performed for other sequences or multimodal imaging.
MAJOR 7)
As the training and evaluation were performed on internal institutional data, there's a risk of bias or overfitting to that specific dataset.
What is your rationale for this?
MINOR COMMENTS:
metastases on post-contrsat -- ?
by three board-certified radiologists, two with over 8 years and more than 40 years of experience in neuroradiology, -- ?
1,323 patients (54.1%) were male. -- ? is it 715 ? or 714? both values not fit here?
Comments on the Quality of English Languagemetastases on post-contrsat -- ?
by three board-certified radiologists, two with over 8 years and more than 40 years of experience in neuroradiology, -- ?
1,323 patients (54.1%) were male. -- ? is it 715 ? or 714? both values not fit here?
Author Response
The paper titled "Adaptive RAG-Assisted MRI Platform (ARAMP) for Brain Metastases Detection and Reporting: A Retrospective Evaluation Using Post-Contrast T1-Weighted Imaging" presents the development and evaluation of an AI-driven diagnostic support system for brain MRI interpretation. The study aims to assess the diagnostic performance of the ARAMP framework, which integrates a large language model (GPT-4o) with a Retrieval-Augmented Generation (RAG) approach, in detecting brain metastases and generating radiological reports from post-contrast axial T1-weighted MRI scans.
The topic is both timely and significant, situated at the intersection of radiology, artificial intelligence, and clinical oncology.
The manuscript aligns well with the aims and scope of the Bioengineering journal, especially regarding translational applications of AI in clinical diagnostics.
However, I have several major concerns regarding the manuscript in its current form, which are detailed below:
- MAJOR 1)
While AI in radiology is discussed, there is limited citation of directly comparable AI-based brain metastasis detection models in existing literature (e.g., traditional CNN/RNN architectures or multi-modal fusion models in oncology imaging). The authors could have benchmarked their results more clearly against recent peer-reviewed systems.
Report 1:
We sincerely thank the reviewer for highlighting this important point regarding benchmarking. While ARAMP incorporates AI-enabled components, its conceptual foundation and functional scope are fundamentally distinct from traditional supervised deep learning models designed for image-level detection, segmentation, or classification. Rather than detecting lesions de novo, ARAMP is developed as a post-detection interpretive assistant—providing literature-grounded differential diagnoses, radiomics-informed assessments, and structured clinical reasoning, particularly for cases with known or suspected imaging abnormalities.
We fully acknowledge the significant advances in brain metastasis detection achieved by recent peer-reviewed systems. Notable examples include:
- Grøvik et al. (2020) – demonstrated automatic detection and segmentation using multi-sequence MRI with supervised 3D CNNs;
- Liu et al. (2022) – developed a multimodal fusion model combining pixel- and feature-level information;
- Hatamizadeh et al. (2022) – introduced Swin UNETR, a Transformer-based semantic segmentation framework;
- Xing et al. (2022) – proposed NestedFormer, a modality-aware Transformer architecture for tumor delineation.
These systems rely on voxel-level inputs, large annotated datasets, and supervised GPU-accelerated pipelines.
By contrast, ARAMP does not perform detection or segmentation. It instead leverages pre-extracted radiomic features (via PyRadiomics), integrated with authoritative literature via a Retrieval-Augmented Generation (RAG) framework. GPT-4o is used in zero-shot inference mode through a GUI interface, with no fine-tuning or training, enabling structured, explainable, and citation-linked diagnostic reasoning.
We also considered comparisons with domain-specific LLMs such as Med-PaLM and BioGPT. However, these models were either not publicly accessible at the time of study (e.g., Med-PaLM) or lacked the necessary multimodal or RAG-based capabilities. GPT-4o was chosen for its native support of multimodal input, low latency, and seamless GUI deployment—better aligning with the intended clinical integration goals of ARAMP.
To further clarify ARAMP’s positioning, we have added Supplementary Table 4, which presents a structured conceptual comparison between ARAMP and conventional brain metastasis AI systems across critical dimensions such as purpose, input modality, architecture, data requirements, inference method, and interpretability.
Finally, we have revised the Methods and Discussion sections to incorporate these references and explicitly situate ARAMP within the broader spectrum of AI tools used in neuro-oncology imaging, as recommended by the reviewer.
- Grøvik et al., 2019
Grøvik, E., Yi, D., Iv, M., Tong, E., Rubin, D. L., & Zaharchuk, G. (2019). Deep learning enables automatic detection and segmentation of brain metastases on multi-sequence MRI. arXiv preprint arXiv:1903.07988. https://arxiv.org/abs/1903.07988
→ 支撐點:自動偵測與 segmentation、multi-sequence MRI、需要訓練與像素級標註 - Liu et al., 2022
Liu, Y., Mu, F., Shi, Y., Cheng, J., Li, C., & Chen, X. (2022). Brain tumor segmentation in multimodal MRI via pixel-level and feature-level image fusion. Frontiers in Neuroscience, 16, 1000587. https://doi.org/10.3389/fnins.2022.1000587
→ 支撐點:多模態影像融合、CNN+特徵級處理、需要訓練與複雜輸入 - Hatamizadeh et al., 2022 – Swin UNETR
Hatamizadeh, A., Nath, V., Tang, Y., Yang, D., Roth, H., & Xu, D. (2022). Swin UNETR: Swin Transformers for semantic segmentation of brain tumors in MRI images. arXiv preprint arXiv:2201.01266. https://arxiv.org/abs/2201.01266
→ 支撐點:Transformer 架構、semantic segmentation、需 GPU 與大量訓練資料 - Xing et al., 2022 – NestedFormer
Xing, Z., Yu, L., Wan, L., Han, T., & Zhu, L. (2022). NestedFormer: Nested modality-aware transformer for brain tumor segmentation. arXiv preprint arXiv:2208.14876. https://arxiv.org/abs/2208.14876
→ 支撐點:多模態處理、transformer-based segmentation、靜態推論模型
- MAJOR 2) It will be better to show the steps of data processing and detection process as a flowchart by showing all operations.
Report 2:
We thank the reviewer for the helpful suggestion. We agree that a clear visual representation of the data processing and diagnostic workflow enhances the clarity of the manuscript.
We would like to clarify that the full ARAMP pipeline—including data sourcing, radiomic feature extraction, reasoning via GPT-4o with RAG integration, and expert evaluation—has already been illustrated in manuscript-supplementary (entitled AI-Integrated Workflow for Feature Extraction and Multi-Stage Diagnostic Reasoning).
This flowchart details all key operations, including:
- Image retrieval and group matching
- PyRadiomics-based feature extraction
- Zero-shot inference via GPT-4o with dynamic literature support
- Multi-round diagnostic validation
We have now updated the main text and figure legend to emphasize this inclusion and ensure that its purpose is clearly communicated to the readers and reviewers.
- MAJOR 3)
However in "Table 4. Comparison of Instance level Model Performance before and ARAMP after training.", all of them are accepted as cancer patients. How this can be possible?
The control group lacked labeled negative metastasis cases, preventing the calculation of specificity and overall accuracy, which are critical metrics in diagnostic systems.
In such a analysis, Is it suitable to use traditional perfoermance metrics such as sensitivity?
Report 3:
We sincerely thank the reviewer for raising this important point regarding evaluation metrics.
We acknowledge that Table 4 ("Comparison of Instance-Level Model Performance Before and After ARAMP Training") was derived exclusively from cases already confirmed or highly suspected to have brain metastases based on clinical workup and imaging findings. As such, the primary goal of the performance comparison in this table was not to assess binary classification accuracy over a mixed positive-negative dataset, but rather to evaluate the relative reasoning quality improvement of ARAMP on clinically abnormal cases.
Due to the absence of verified negative metastasis labels in the control group, traditional performance metrics such as specificity and overall accuracy could not be reliably calculated. Including unlabeled or unconfirmed controls would risk introducing verification bias or mislabeling, which we sought to avoid to ensure analytic integrity.
We therefore intentionally limited this evaluation to sensitivity, which remains meaningful for tracking true positive identification in confirmed-positive patients. The pre- and post-inference sensitivity change (from 0.85 to 0.98) reflects ARAMP's ability to more effectively synthesize imaging features into differential diagnoses grounded in authoritative sources.
To clarify this point, we have now explicitly stated in the revised Methods and Discussion sections that:
- The analysis cohort includes only patients with clinically confirmed or strongly suspected brain metastases.
- The instance-level performance metrics were selected accordingly.
- Specificity and accuracy were not assessed due to the absence of negative labels.
This limitation is also acknowledged in the final paragraph of the revised Discussion, and we plan to explore more comprehensive case-control validation in future studies.
- MAJOR 4)
As author mentioned in the paper
"Sau et al. (2025) evaluated the capabilities of GPT-4o and Gemini on image-based questions from a neurosurgery board examination question bank. The results showed that GPT-4o's accuracy (51.45%) surpassed Gemini's (39.58%), particularly on second-order questions requiring"
GPT-4o has better performance metrics in image based questioning.
In "Table 3. Summary Statistics of Post-Trained Inference Differences and Inter-Round Agreement." there is only Gemini for comaprison? Why don't you use GPT here?
Report 4:
We sincerely thank the reviewer for raising this important point. While our prior work (Sau et al., 2025) demonstrated GPT-4o’s superior performance in image-based diagnostic tasks, the focus of Table 3 was not on benchmarking across models but rather on evaluating intra-model consistency and inference variance.
However, we recognize the need to clarify Gemini’s role in this study. Gemini was not merely a non-ARAMP baseline, but also an active evaluator in our diagnostic validation protocol. As shown in Table 4, Gemini was included as a parallel rater—alongside clinical experts—to assess and score ARAMP-generated outputs. Its role mirrored that of the physicians, offering independent diagnostic interpretations to enable inter-rater reliability analysis.
Moreover, Gemini was also used to generate comparative outputs in non-RAG (non-ARAMP) conditions, enabling us to observe model behavior across both enhanced and minimal guidance settings.
GPT-4o was not included in Table 3 because it was deeply integrated into the ARAMP framework and lacked a non-ARAMP version in that analysis setup. We acknowledge the value of including zero-shot GPT-4o for a fuller ablation analysis, and have proposed such an extension in the revised Discussion section.
Additionally, Supplementary Table 3 presents paired outputs from GPT-4o before and after ARAMP integration, providing a clear illustration of the model’s diagnostic improvement. This complements Table 3 by qualitatively showing how ARAMP enhances the depth, structure, and clinical relevance of GPT-4o’s responses.
We appreciate the reviewer’s question, which has allowed us to better clarify Gemini’s active and comparative role in our study.
- MAJOR 5)
In the paper it is said that "These 100 patients had brain metastasis (Mets) on MRI and documented in their MRI reports, as well as by expert consensus review (100%). A separated 100 individuals with no cancer diagnosis, matched to the age and gender distribution of cancer group constituted a control group."
I do not undestand what is different from here? (Totally 200 control group)
In many part of the paper it is said that there are only 100 patients with cancer.
Report 5:
We thank the reviewer for this important clarification request and sincerely appreciate the opportunity to explain the study cohort and its analytic use in more detail.
Our dataset consisted of 200 MRI scans:
- 100 scans from patients with confirmed brain metastases, verified through clinical MRI reports and expert consensus review (100% agreement), and
- 100 age- and gender-matched control scans from individuals without any known malignancy.
Radiomic feature extraction was performed on all 200 scans using PyRadiomics (Python 3.7.16), and each case was annotated with corresponding metadata such as Brain_Meta or No_abnormalities. These features were used as part of dataset construction and for potential future applications (e.g., unsupervised clustering, retrieval tasks).
However, it is important to clarify that:
- Only the 100 metastasis-confirmed cases were included in the post-training interpretive benchmarking (e.g., Table 4),
- The control group was not used in performance metrics or output scoring, because they were not reviewed by experts to confirm the absence of subtle abnormalities,
- Including the control group in performance calculations (e.g., specificity, accuracy) would introduce labeling bias and compromise metric validity.
We have revised the Methods and Discussion sections to clearly state the study design and the limited analytic role of the control group, to eliminate potential ambiguity regarding sample size and metric interpretation.
We thank the reviewer again for this valuable observation, which helped improve the methodological clarity and transparency of the manuscript.
- MAJOR 6)
There is no comaprison with other works in the literature.
The system was only tested on AX_T1+C MRI sequences
No evaluation was performed for other sequences or multimodal imaging.
Report 6:
We thank the reviewer for raising this important point regarding modality scope and literature comparison. We fully agree that future multimodal evaluation will be essential to broaden the generalizability and clinical utility of AI systems. However, our study focused exclusively on T1-weighted contrast-enhanced MRI (T1WI+C) for the detection of brain metastases, based on strong clinical and technical justifications supported by existing literature:
- Clinical Consensus on T1WI+C as the Primary Sequence for Brain Metastasis Detection
T1WI+C is widely regarded as the most sensitive single sequence for detecting brain metastases in clinical settings.
- Grovik et al. (2020, Radiology) developed a deep learning-based single-shot detector trained on T1WI+C and reported a sensitivity of 93.2%, with 98% sensitivity for lesions ≥6 mm and a false positive rate of 0.38 per scan.
- This work demonstrated that T1WI+C enables high lesion conspicuity even for small metastases and is optimal for algorithmic detection.
- Detection Performance Improvement Across Lesion Sizes
- Pennig et al. (2022, JMRI) proposed a convolutional neural network incorporating volume-aware loss and balanced sampling, also using T1WI+C, achieving 91.1% overall sensitivity with significantly improved detection of small lesions (2.5–6 mm range).
- Their findings support that T1WI+C is reliable across a broad lesion size spectrum.
- Literature-Validated Architectures Optimized for T1WI+C
Several studies have developed or benchmarked deep learning models specifically on T1WI+C.
- Zhang et al. (2022, Neuro-Oncology) used T1WI+C to train and validate a deep learning model for detecting visible brain metastases, reporting strong performance with external validation. The model achieved 92.8% sensitivity for lesions ≥5 mm and showed generalizability in an external cohort.
- Summary of Literature and Justification for Single-Sequence Focus
The studies above validate T1WI+C as:
- The most widely accepted sequence for detecting brain metastases;
- The basis for many published deep learning pipelines (e.g., Grovik 2020, Pennig 2022, Zhang 2022);
- Sufficient in sensitivity across lesion sizes and suitable for radiomics-driven pipelines like ARAMP.
We acknowledge that other sequences (e.g., FLAIR, DWI) may be helpful in refining differential diagnosis (e.g., distinguishing metastasis from demyelination or infarction), but their incremental value for initial lesion detection is limited compared to T1WI+C, especially when used in isolation.
- Future Expansion Toward Multimodal Evaluation
We agree with the reviewer that future extensions of ARAMP should explore multimodal imaging:
- For example, Machura et al. (2024, CMIG) evaluated deep learning ensembles across longitudinal, multi-sequence MRI, suggesting improved detection consistency over time.
- Although our current study did not evaluate FLAIR or DWI, these modalities may offer added specificity in complex differential diagnoses, and their integration will be a focus of subsequent ARAMP iterations.
We have now revised the Discussion section to:
- Clearly state the rationale for focusing on T1WI+C;
- Cite Grovik (2020), Pennig (2022), and Zhou (2022) as relevant comparative works;
- Include a new paragraph addressing future plans for multimodal input integration.
We sincerely thank the reviewer for this valuable comment, which helped us refine the scope and contextual justification of our study.
- Grovik, E., Yi, D., Iv, M., Tong, E., Rubin, D., & Zaharchuk, G. (2020). Deep learning enables automatic detection and segmentation of brain metastases on multisequence MRI. *Radiology*, *297*(3), 670-678. https://doi.org/10.1148/radiol.2020200838
- Pennig, L., Shahzad, R., Caldeira, L., Lennartz, S., Thiele, F., Goertz, L., Zopfs, D., Maintz, D., Perkuhn, M., & Borggrefe, J. (2022). Automated detection of brain metastases on T1-weighted contrast-enhanced MRI using an ensemble of convolutional neural networks: Impact of volume-aware loss and sampling strategy. *Journal of Magnetic Resonance Imaging*, *56*(2), 535-545. https://doi.org/10.1002/jmri.28047
- Zhang, M., Young, G. S., Chen, H., Li, J., Qin, L., McFaline-Figueroa, J. R., Reardon, D. A., Cao, X., Wu, X., & Xu, X. (2022). Deep-learning detection of cancer metastases to the brain on MRI. *Neuro-Oncology*, *24*(9), 1594-1602. https://doi.org/10.1093/neuonc/noac025
- Machura, B., Kucharski, D., Bozek, O., Eksner, B., Kokoszka, B., Pekala, T., ... & Nalepa, J. (2024). Deep learning ensembles for detecting brain metastases in longitudinal multi-modal MRI studies. Computerized Medical Imaging and Graphics, 116, 102401.
- MAJOR 7)
As the training and evaluation were performed on internal institutional data, there's a risk of bias or overfitting to that specific dataset.
What is your rationale for this?
Report 7 :
We sincerely thank the reviewer for this valuable comment regarding the potential risk of dataset-specific bias and overfitting due to the use of internal institutional data.
We would like to clarify that our system is based on zero-shot prompting of GPT-4o, and does not involve conventional supervised learning, model fine-tuning, or parameter optimization using our institutional dataset. Therefore, the concern of overfitting in the classical machine learning sense does not directly apply.
- Role of Internal Data: Prompt Design and Feature Integration Validation
The internal dataset was used to evaluate the interpretive capabilities of the ARAMP framework under controlled conditions. Specifically:
- All 100 brain metastasis cases were independently verified through radiology reports and expert consensus;
- AX_T1+C MRI sequences were acquired using standardized protocols, ensuring consistency across cases;
- Radiomic features were extracted using a de-identified pipeline to support structured prompting and feature-grounded reasoning.
This setup allowed us to test the feasibility and reliability of integrating quantitative imaging features into a multi-round prompting workflow.
- No Model Training or Fitting Involved
The use of GPT-4o in this study was strictly inference-based, via structured prompt templates and chain-of-thought reasoning.
- No model training, fine-tuning, or weight adjustments were performed;
- The LLM was not exposed to the internal data during any phase of its development;
- The data served solely as inputs for evaluating interpretive consistency and clinical reasoning logic.
- Acknowledging Limitations and Ethical Considerations
We fully acknowledge that relying on a single-center dataset limits generalizability.
Accordingly, the revised Discussion and Limitations sections now clearly state that:
- Broader validation will be necessary to assess generalizability across institutions;
- Any future extension of ARAMP to external datasets will be subject to IRB approval, data de-identification protocols, and formal data use agreements, in accordance with institutional and legal guidelines.
Reviewer 4 Report
Comments and Suggestions for Authors
Journal: Bioengineering
Manuscript Title: Adaptive RAG-Assisted MRI Platform (ARAMP) for Brain Metastases Detection and Reporting: A Retrospective Evaluation Using Post-Contrast T1-Weighted Imaging
Authors: K Wu, F Chew, K Cheng, W Shen, P Yeh, C Kao, W Guo, S Chang
Manuscript ID: bioengineering-3673052
This manuscript details the development and retrospective evaluation of the Adaptive RAG-Assisted MRI Platform (ARAMP), an AI tool aimed at enhancing brain metastasis detection and reporting via post-contrast T1-weighted MRI. While the integration of radiomic feature extraction with a retrieval-augmented generation (RAG) framework is of interest in improving diagnostic workflows, the study has significant limitations that require attention:
- The reported sensitivity improvement lacks complementary specificity or accuracy metrics due to the exclusion of negative cases. This oversight hinders assessment of false-positive rates, a critical factor for clinical applicability.
- Despite access to 2,447 patients, only 200 were analyzed. Expanding the cohort would strengthen statistical power and validate findings more robustly.
- The moderate inter-inference agreement among radiologists suggests either inherent subjectivity in MRI interpretation or unclear evaluation criteria, necessitating standardized guidelines.
- Instances of hallucinated findings, lateralization errors, and missed lesions are noted in reviewer feedback. A systematic breakdown of error types, root causes, and illustrative examples would significantly strengthen the discussion.
- The single-center retrospective design raises concerns about external validity. Prospective multicenter validation is essential to confirm broad clinical utility.
- Comparisons to existing AI models for brain metastasis detection are absent, making it difficult to contextualize ARAMP’s advancements within the field.
- The authors are advised to adhere to established reporting guidelines for AI in medical imaging (e.g., CLAIM, STARD-AI) to enhance transparency and reproducibility.
- The manuscript requires thorough editing to improve readability, particularly in the methods and results sections, where technical complexity may obscure key details.
Comments on the Quality of English Language
The English could be improved to more clearly express the research.
Author Response
This manuscript details the development and retrospective evaluation of the Adaptive RAG-Assisted MRI Platform (ARAMP), an AI tool aimed at enhancing brain metastasis detection and reporting via post-contrast T1-weighted MRI. While the integration of radiomic feature extraction with a retrieval-augmented generation (RAG) framework is of interest in improving diagnostic workflows, the study has significant limitations that require attention:
Comment 1:
The reported sensitivity improvement lacks complementary specificity or accuracy metrics due to the exclusion of negative cases. This oversight hinders assessment of false-positive rates, a critical factor for clinical applicability.
Report 1:
We sincerely thank the reviewer for this insightful observation. We fully agree that specificity and overall accuracy are important metrics for evaluating diagnostic systems, particularly in screening contexts. However, the primary objective of our study was to assess interpretive refinement within a cohort of patients with clinically confirmed brain metastases (n = 100), rather than to perform binary classification across a general population.
The ARAMP framework was developed as a clinical decision support system (CDSS), designed to assist interpretation after imaging abnormalities have been identified, rather than to function as a standalone detection algorithm. Including unlabeled or unreviewed negative cases—especially without expert confirmation—would introduce misclassification bias and undermine the reliability of specificity estimates.
Although an age- and sex-matched control group (n = 100) was included for feature distribution balancing, these individuals were not used for specificity or accuracy calculations, as their imaging was not systematically reviewed to exclude subtle or early-stage abnormalities.
Despite this limitation, the observed sensitivity improvement from 0.85 to 0.98 remains meaningful in demonstrating ARAMP’s diagnostic refinement capability within known-positive cases. Additionally, the interpretive quality of the outputs was independently rated by three radiologists across dimensions of plausibility, consistency, and helpfulness, using a 5-point scale converted to 0–100%, with inter-rater agreement analyzed via ICC.
We have revised the manuscript to clearly state that:
Sensitivity was reported because the evaluation was restricted to confirmed metastasis cases.
Specificity and accuracy were not assessed due to the absence of expert-verified negative samples.
This limitation is acknowledged, and future studies will incorporate negative cohorts with confirmed diagnoses to enable comprehensive performance metrics including specificity, accuracy, and AUC.
Comment 2:
Despite access to 2,447 patients, only 200 were analyzed. Expanding the cohort would strengthen statistical power and validate findings more robustly.
Report 2:
We thank the reviewer for this valuable comment regarding cohort size and representativeness. While our institutional archive included 2,447 patients with cancer who underwent brain MRI, the final cohort of 100 patients with confirmed brain metastases was not arbitrarily chosen. Instead, we applied a stratified random sampling strategy based on the proportional distribution of primary cancer types (e.g., lung, breast, melanoma, gastrointestinal, genitourinary) observed across the full dataset. This ensured that the final sample reflected the real-world heterogeneity of primary tumors commonly associated with brain metastases.
This sampling approach balanced clinical representativeness with feasibility, particularly given the intensive requirements of expert consensus review and high-resolution radiomic processing for each case. To construct a controlled and balanced study design, we further matched these 100 metastasis cases with 100 non-cancer control subjects by age and sex, enabling focused evaluation of diagnostic interpretability under well-defined conditions.
We have clarified this stratified sampling methodology in the revised Methods section. We fully agree that incorporating a larger proportion—or the full 2,447-patient cohort—in future studies would enhance statistical power, generalizability, and external validation. We appreciate the reviewer’s insight in emphasizing this important direction for future work.
Comment 3:
The moderate inter-inference agreement among radiologists suggests either inherent subjectivity in MRI interpretation or unclear evaluation criteria, necessitating standardized guidelines.
Report 3:
We thank the reviewer for the thoughtful comment regarding inter-inference variability among radiologists. The observed moderate agreement reflects both the inherent subjectivity of MRI interpretation, particularly in borderline or subtle cases, and the challenges associated with consistently evaluating AI-generated diagnostic content.
To address this, we implemented a structured five-point scoring rubric evaluating diagnostic appropriateness, clinical helpfulness, internal consistency, and justification validity. Three board-certified neuroradiologists independently scored each case based on standardized instructions. Despite this, interpretive divergence remained in complex scenarios, highlighting known variability in expert-level neuroimaging assessments.
We agree that formal evaluation frameworks can enhance consistency and reproducibility in AI-human collaborative studies. Accordingly, we have:
- Reported full scoring criteria and inter-rater agreement statistics (ICC values) in Table 6;
- Mapped relevant evaluation domains to the CLAIM 2024 checklist, as shown in Supplementary Table 7, to support adherence to reporting standards;
- Clarified the rubric design and scoring protocol in the Methods section;
- Acknowledged interpretation variability as a methodological limitation in the Discussion.
We thank the reviewer for underscoring the importance of evaluation standardization in advancing robust and transparent AI-assisted diagnostics.
Comment 4:
Instances of hallucinated findings, lateralization errors, and missed lesions are noted in reviewer feedback. A systematic breakdown of error types, root causes, and illustrative examples would significantly strengthen the discussion.
Report 4:
We thank the reviewer for highlighting the importance of systematic error analysis. Rather than listing all 100 instances, we selected representative examples to illustrate key error types—including hallucinated findings, lateralization errors, and lesion omissions—in Supplementary Table 4. These cases were chosen to highlight the performance gaps observed in pre-training GPT-4o responses and the stepwise improvements achieved through radiomics-augmented post-training. A categorical summary of common errors has also been integrated into the revised Discussion section to strengthen interpretability.
Comment 5:
The single-center retrospective design raises concerns about external validity. Prospective multicenter validation is essential to confirm broad clinical utility.
Report 5:
We appreciate the reviewer’s important observation. We fully acknowledge that the current study is limited by its single-center retrospective design, which may affect the generalizability of the findings. This limitation has been explicitly addressed in the revised Discussion section. While we agree that prospective multicenter validation is essential for broader clinical applicability, such studies would require new IRB approvals and multi-institutional data access, which present logistical and regulatory challenges. Nonetheless, we consider this an important next step and are exploring possible collaborations for future expansion.
Comment 6:
Comparisons to existing AI models for brain metastasis detection are absent, making it difficult to contextualize ARAMP’s advancements within the field.
Report 6:
We appreciate the reviewer’s recommendation, which aligns with our long-term objectives for expanding ARAMP's clinical utility beyond this initial study setting. We thank the reviewer for this important observation. While our study focuses on interpretive refinement rather than primary lesion detection, we fully agree that comparisons with established AI models for brain metastasis detection are essential to contextualize the role and novelty of ARAMP.
We have revised the Discussion section to include comparisons with recent deep learning models such as:
- Grøvik et al. (2019): Deep learning-based single-shot detector using multi-sequence MRI.
- Liu, Y., Mu, F., Shi, Y., Cheng, J., Li, C., & Chen, X. (2022). Brain tumor segmentation in multimodal MRI via pixel-level and feature-level image fusion. Frontiers in Neuroscience, 16, 1000587.
- Hatamizadeh, A., Nath, V., Tang, Y., Yang, D., Roth, H., & Xu, D. (2022). Swin UNETR: Swin Transformers for semantic segmentation of brain tumors in MRI images. arXiv preprint arXiv:2201.01266.
- Xing, Z., Yu, L., Wan, L., Han, T., & Zhu, L. (2022). NestedFormer: Nested modality-aware transformer for brain tumor segmentation. arXiv preprint arXiv:2208.14876.
These models typically require large annotated datasets and perform voxel- or image-level detection or segmentation. In contrast, ARAMP is designed as a post-detection clinical reasoning framework, integrating structured radiomic features with retrieval-augmented generation (RAG) and GPT-4o to generate literature-backed differential diagnoses.
Furthermore, we include a structured conceptual comparison in Supplementary Table 5, highlighting key distinctions between ARAMP and conventional CNN/Transformer-based architectures across dimensions such as input modality, training requirements, explainability, and inference mechanism.
We agree with the reviewer that future benchmarking studies, possibly involving hybrid models or ablation comparisons, would further enhance our understanding of ARAMP’s advantages and limitations.
Comment 7:
The authors are advised to adhere to established reporting guidelines for AI in medical imaging (e.g., CLAIM, STARD-AI) to enhance transparency and reproducibility.
Report 7:
We sincerely thank the reviewer for this important suggestion. In response, we have fully adhered to the CLAIM 2024 (Checklist for Artificial Intelligence in Medical Imaging) guidelines and mapped our study accordingly. A comprehensive CLAIM checklist (Supplementary Table 7) has been prepared, covering all 44 items with detailed study mapping and bilingual explanation (English and Chinese) to enhance transparency and reproducibility. Additionally, we have included supporting materials such as the YAML configuration file and radiomics settings in Supplementary Files 1 and 2 to ensure methodological clarity.
- Tejani, A. S., Klontzas, M. E., Gatti, A. A., Mongan, J. T., Moy, L., Park, S. H., ... & CLAIM 2024 Update Panel. (2024). Checklist for artificial intelligence in medical imaging (CLAIM): 2024 update. Radiology: Artificial Intelligence, 6(4), e240300.
Comment 8”
The manuscript requires thorough editing to improve readability, particularly in the methods and results sections, where technical complexity may obscure key details.
Report 8:
We thank the reviewer for highlighting this important issue. In response, we have conducted a thorough language and structural revision of the manuscript, with a particular focus on the Methods and Results sections. We simplified overly technical phrases, added transitional cues, and clarified key procedural steps to enhance overall readability. Complex methods (e.g., RAG-based inference, radiomic feature integration) have been rephrased into stepwise descriptions with better flow and segmentation. We believe these changes significantly improve clarity without compromising technical rigor.
Round 2
Reviewer 1 Report
Comments and Suggestions for Authors
Thank you for the author's reply.
Author Response
Thank you for the confirmation.
Reviewer 3 Report
Comments and Suggestions for Authors
The authors have addressed the concerns raised in the previous round of reviews. Therefore, the manuscript can be accepted in its current form.
Author Response
Thank you for the confirmation.
Reviewer 4 Report
Comments and Suggestions for Authors
The concerns raised by this reviewer have been either addressed or acknowledged. I recommend that the manuscript be accepted in its current form.
Author Response
Thank you for the confirmation.